# Good Cop, Bad Cop: Defining the Roles of Δ40p53 in Cancer and Aging

**DOI:** 10.3390/cancers12061659

**Published:** 2020-06-23

**Authors:** Luiza Steffens Reinhardt, Xiajie Zhang, Anna Wawruszak, Kira Groen, Geoffry N. De Iuliis, Kelly A. Avery-Kiejda

**Affiliations:** 1Priority Research Centre for Cancer Research, Innovation and Translation, School of Biomedical Sciences and Pharmacy, Faculty of Health and Medicine, University of Newcastle, Newcastle, NSW 2305, Australia; Luiza.SteffensReinhardt@uon.edu.au (L.S.R.); xiajie.zhang@newcastle.edu.au (X.Z.); awawruszak@gmail.com (A.W.); Kira.Groen@newcastle.edu.au (K.G.); 2Hunter Cancer Research Alliance, Newcastle, NSW 2305, Australia; 3Hunter Medical Research Institute, Newcastle, NSW 2305, Australia; geoffry.deiuliis@newcastle.edu.au; 4Department of Biochemistry and Molecular Biology, Medical University of Lublin, 20-093 Lublin, Poland; 5Priority Research Centre for Reproductive Science, School of Environmental and Life Sciences, Faculty of Science, University of Newcastle, Callaghan, Newcastle, NSW 2308, Australia

**Keywords:** p53 isoforms, Δ40p53, p53, cancer, aging

## Abstract

The tumour suppressor p53 is essential for maintaining DNA integrity, and plays a major role in cellular senescence and aging. Understanding the mechanisms that contribute to p53 dysfunction can uncover novel possibilities for improving cancer therapies and diagnosis, as well as cognitive decline associated with aging. In recent years, the complexity of p53 signalling has become increasingly apparent owing to the discovery of the p53 isoforms. These isoforms play important roles in regulating cell growth and turnover in response to different stressors, depending on the cellular context. In this review, we focus on Δ40p53, an N-terminally truncated p53 isoform. Δ40p53 can alter p53 target gene expression in both a positive and negative manner, modulating the biological outcome of p53 activation; it also functions independently of p53. Therefore, proper control of the Δ40p53: p53 ratio is essential for normal cell growth, aging, and responses to cancer therapy. Defining the contexts and the mechanisms by which Δ40p53 behaves as a “good cop or bad cop” is critical if we are to target this isoform therapeutically.

## 1. Introduction

DNA is the repository of all genetic information within a cell and it is under constant threat of damage from a variety of environmental and endogenous sources [1]. Defects in DNA repair are catastrophic. At a cellular level, the accumulation of DNA damage can lead to chromosome loss and/or abnormalities and cell death; which at an organismal level can give rise to embryonic lethality and organ/tissue malformation as well as neurological damage, premature aging, and cancer [2,3]. Hence, the ability to repair damaged DNA with high fidelity is essential for normal eukaryotic cell development as well as maintaining tissue homeostasis. 

p53 has been widely described as the “guardian of the genome” due to its ability to coordinate multiple and diverse signalling pathways involved in cellular homeostasis and the DNA damage response [4,5,6,7,8]. p53 facilitates the repair of DNA through the induction of cell cycle arrest, and when DNA damage is severe, it induces apoptosis [9,10,11]. Critical to its function as a tumour suppressor, p53 has been shown to suppress inflammation through a number of different mechanisms including the suppression of excessive production of reactive oxygen species and the inhibition of Nuclear Factor-κB (NF-κB) [12]. p53 also plays a major role in cellular senescence and aging [13]. 

A seminal discovery in 2005 revealed the existence of several p53 isoforms that have distinct functions [14]. One of these isoforms, Δ40p53 (also known as p47 and ΔNp53), is an N-terminal truncation of full-length p53 (FLp53) that lacks the first transcriptional activation domain (TAD I). Initial reports suggested that Δ40p53 had no activity on its own and that it acted as a dominant-negative regulator of p53-mediated transactivation, growth suppression, and apoptosis [15,16]. It is now apparent that Δ40p53’s functions are far more complex. Δ40p53 can alter p53 target gene expression in both a positive and negative manner [16,17,18]—as has been described for the full-length protein [19]—and this is largely dependent on the Δ40p53:FLp53 ratio and on the cellular context [20]. Because of this, it is still unclear whether Δ40p53 is able to sustain the tumour suppressive properties of FLp53 and whether its expression is beneficial in cancer. In addition, a vital role in embryonic stem cell regulation [21] and aging has also been described for Δ40p53 in mice [22,23], further showing the critical involvement of this isoform in both embryonic and somatic cells. However, most of the research conducted to date in attempts to decipher the function of this isoform have utilised overexpression models, and very little is still known regarding C-terminal variants of the Δ40p53 isoform, Δ40p53β, and Δ40p53γ. In this review, we will summarise what is currently known about Δ40p53 in cancer and aging, whilst highlighting the contexts where it behaves as a “good” or “bad” modulator of p53 function.

## 2. The Canonical p53 Pathway

DNA integrity is essential for sustaining cellular functions. Any mutations or alterations in genes critical for the maintenance of genomic stability increase the susceptibility to diseases, such as cancer [24]. The importance of p53 in tumour suppression has long been established with somatic *TP53* mutations present in about 50% of all human cancers [4,25]. The role of p53 as a transcriptional activator has been extensively reviewed elsewhere [13,26,27] and will only briefly be discussed here. The p53 protein remains at extremely low expression levels in normal cells through regulatory control conferred by a ring finger type E3 ubiquitin ligase named Mouse Double Minute-2 (MDM2) or Human Double Minute-2 (HDM2) in humans, which binds specifically to TAD I of the p53 protein, ubiquitinating p53 for degradation [28,29,30]. Additionally, the ARF tumour suppressor (alternative reading frame, protein product of INK4a locus, p14^ARF^ in human and p19^ARF^ in mice) is another key regulator of p53 that is activated by oncogenic signals and binds directly to HDM2, thus inhibiting p53 degradation [31,32,33].

A wide variety of stressors, such as DNA damage, oxidative stress, or hypoxia lead to the accumulation of the activated p53 tetramer, as detailed in Figure 1 [34,35,36]. HDM2 becomes sumoylated, facilitating self-degradation, and p53 therefore becomes stabilized and accumulates in the nucleus [37]. This allows p53 to bind to DNA motifs known as p53 response elements (p53RE), in the promoter region of target genes to initiate either target gene activation or repression [38,39]. Under low-level stress, such as low concentrations of DNA-damaging agents, p53 activation initiates cellular functions involved in preserving cell survival and maintaining genomic stability, such as cell cycle arrest or DNA repair [35,40]. In response to potent stressors, p53 transactivates a network of genes such as *BAX* (BCL2-associated X), *NOXA* (phorbol-12-myristate-13-acetate-induced protein 1) and *PUMA* (p53 upregulated modulator of apoptosis), which initiate apoptosis or senescence of severely damaged cells [34,41,42,43,44,45]. A recent census of p53 target genes indicates that p53 binding to its RE is independent of cell type and treatment. p53 primarily acts as a direct activator of transcription, whereas, downregulation of its target genes occurs through an indirect mechanism and requires p21 [46]. In this context, even though it is known that p53 targets different genes depending on the stress intensity, the molecular mechanisms behind this process still remains to be clarified [47,48,49]. 

Persistent oncogenic signals, as well as the accumulation of oxidative damage by reactive oxygen species (ROS) are key determinants of senescence that eventually result in aging and neurodegenerative diseases [13]. In this regard, p53 has well known roles in regulating cell metabolism and survival through inhibition of the IGF-1/AKT and mTOR pathway. Stress-activated p53 suppresses these two pathways by activating IGF-BP3, PTEN, and Tsc2, allowing replication errors to be corrected [50]. Therefore, the ability of p53 to act as a transcriptional activator or repressor of wide variety of genes allows it to orchestrate the increasingly complex decisions of cell fate in response to genomic stress [34,35,40].

## 3. p53 Isoforms

The identification of smaller p53 isoforms has brought further diversity and complexity to the understanding of p53 signalling. FLp53 contains three functional domains: TAD I and II, a DNA-binding domain (DBD) and an oligomerization domain (OD) (Figure 2A); which are critical for p53 to form a functional tetramer, accurately recognize its DNA binding sequence and successfully initiate transcription of target genes. However, N-terminally truncated isoforms lack one (TAD I, Δ40p53) or both (TAD I and II, Δ133p53 and Δ160p53) TADs and can be generated through alternative splicing (Δ40p53), alternative promoter usage (Δ133p53 and Δ160p53) and alternative initiation of translation at ATG40 or ATG160 (Δ40p53 and Δ160p53). Additionally, C-terminally truncated p53 isoforms, p53β or p53γ, are generated through alternative splicing of intron 9. The N-terminal truncations can co-exist with the C-terminal variants, resulting in at least 12 protein isoforms: p53, p53β, p53γ, ∆40p53α, ∆40p53β, ∆40p53γ, ∆133p53α, ∆133p53β, ∆133p53γ, ∆160p53α, ∆160p53β, and ∆160p53γ [13,42,43,44]. Another two isoforms Δp53 and p53ψ have also been described [45,46]. Given the contradictory roles that have been described for Δ40p53 in processes such as cancer and aging, the remainder of this review will focus on this isoform, in an effort to elucidate the mechanisms by which it can positively or negatively regulate FLp53.

## 4. The ∆40p53 Isoform

### 4.1. Mechanism of Production

Δ40p53 is a natural isoform of FLp53 that is present in normal cells under normal growth conditions where it can be produced by alternative splicing, alternative translation or post translational modulation. 

Δ40p53 can arise through alternative splicing, which retains intron 2, resulting in three in-frame stop codons (Figure 2A). This prevents translation of FLp53 and the main protein product generated is Δ40p53 [51]. This mechanism of Δ40p53 production was reported in breast cancer cells [51]. The shift from fully-spliced mRNA to mRNA retaining intron 2 is associated with guanine-rich sequences in intron 3 of p53 pre-mRNA. Four 5’-monophosphates (GMPs) are prone to form a G-tract via Hoogsteen hydrogen-bonds with the help of the monovalent cation potassium ion (K^+^) [52]. G-tracts stack on each other and form 4-stranded DNA or RNA structures called G-quadruplex (G4), which are highly stable. G4 affects splicing when present in introns or translation when present in mRNA [52,53]. G4 in intron 3 may regulate the splicing of intron 2 given that exon 3 is very short [52]. In addition, a polymorphism rs17878362 (*TP53* PIN3) in intron 3, lying in close vicinity of G4, whose duplication increased the distance between G4 and intron 2, is associated with increased cancer risk [54,55] and penetrance of germline mutations in p53 [56]. The PIN3 wild-type (wt) allele results in a fully-spliced p53 mRNA, subsequently altering the ratio of Δ40p53:FLp53; moreover, wtPIN3 homozygotes have lower cancer risk than homozygotes with the polymorphic allele [52,53,55,57]. The relationship between PIN3 and another polymorphism rs1642785 (PIN2) located in intron 2 is elusive. However, the combination of homozygous wtPIN2 and wtPIN3 may be optimal for the production of FLp53 [53].

Δ40p53 can also be produced by alternative translation from the start codon at AUG40 of the proximal promoter of the p53 gene (Figure 2A) [15,51]. The eukaryotic initiation factors (eIFs) are decreased under cellular stresses, affecting the formation of translation-initiation complex at the cap site, thus affecting global protein synthesis [16,58]. p53 mRNA includes two internal ribosome entry sites (IRESs), and this is critical in regulating FLp53 and Δ40p53 translation in response to stress [59,60]; however, different drug treatments selectively favour one or the other IRES. A study looking at the cap-independent mechanism of Δ40p53/p53 protein synthesis has found that induction of endoplasmic reticulum (ER) stress, using tunicamycin and thapsigargin, upregulated the expression of both Δ40p53 and p53—with the former to the same extent by both drugs and the latter less induced by thapsigargin than by tunicamycin. Doxorubicin had no effect on the expression of either isoform [61].

More recently, post-translational degradation of the FLp53 protein via the 20S proteasome was shown to generate Δ40p53 [62], adding to the above mechanisms generating Δ40p53. Δ40p53 formed heterotetramers with FLp53 and attenuated its transcriptional activity in a dominant negative manner. Additionally, the generation of the Δ40p53 isoform by proteolytic cleavage was enhanced under oxidative stress, suggesting that this may represent a negative feedback loop by which to keep FLp53 activities in check [62]. It is unknown whether the Δ40p53 protein product of proteasome cleavage is similar to that previously described to be produced by calpain cleavage of FLp53 [63] and additional identification methods, such as mass spectrometry of the calpain cleavage product, are necessary to fully characterise whether these are the same two protein products. Nevertheless, this represents an exciting new avenue for the production of the Δ40p53 isoform and further shows the complexity of the p53 pathway. It also emphasises the critical role played by Δ40p53, an isoform produced through multiple mechanisms to regulate FLp53 functional activities; this level of complexity has so far not been described for any of the other p53 isoforms.

### 4.2. The In Vitro Function of ∆40p53

Most of our knowledge on Δ40p53 function comes from studies performed in cancer cell lines [15,16,20,51,64,65,66]. The function of Δ40p53 in various types of cancer cells in vitro is summarised in Table 1 and Figure 2B and Figure 3; and will be discussed in detail in the following sections.

#### 4.2.1. Modulation of Transcription of p53-Target Genes

In melanoma cells, Δ40p53 has been shown to repress *p21^WAF1^* and *PUMA* expression when expressed in excess to FLp53, both at the basal level or following treatment with DNA damaging agents. However, Δ40p53 had no transcriptional activity in the absence of FLp53 on these genes [5,15,16]. The retention of TAD II suggests that Δ40p53 can transcriptionally control a subset of p53 target genes [70]. This was confirmed by in vitro studies in the p53-null H1299 human lung-carcinoma cell line where high Δ40p53:FLp53 expression resulted in the upregulation of *BAX* and pro-apoptotic activities, contradicting previous findings that Δ40p53 exhibited only dominant-negative effects on FLp53 [15,17]. In concordance, Ota et al. observed increased expression of *p21^WAF1^*, *IL-8*, *HDM2* and *FAS* in response to ectopic Δ40p53 in *TP53* null hepatocellular carcinoma (HCC) cells [64]. In addition, the level of Δ40p53 expression, relative to FLp53, mediated the transcription of p53 target genes in response to the cytotoxic agent LA 12 in breast cancer cells [65]. Furthermore, in vitro studies in melanoma have also revealed that high expression of Δ40p53 corresponded to increased activity of endogenous FLp53 accompanied by increased expression of the apoptotic gene *PIDD* [66].

A study by Hafsi et al. showed that Δ40p53 had no transcriptional activity and that its effects on FLp53 were cell line dependent. Δ40p53 activity was shown to increase under genotoxic stress and mediate p53-inducible target genes, including upregulation of *p21^WAF1^*, to a greater degree than FLp53 alone [20]. This reinforces the importance of the Δ40p53:FLp53 ratio in controlling the p53 response, and demonstrates one mechanism whereby Δ40p53 can exert dominance over FLp53 activity [18,20]. Accordingly, the Δ40p53:FLp53 ratio was shown to be important in the regulatory function that Δ40p53 has towards FLp53. Δ40p53 was inhibitory when expressed in excess of FLp53 (≥3-fold). However, when Δ40p53 was expressed at a lower or equal level to FLp53, Δ40p53 had more complex effects (from inhibition to activation), depending on the cellular context. Δ40p53 suppressed FLp53 transcriptional activity in an incremental manner, depending on the dosage of Δ40p53 in H1299 cells. However, inhibition of p53 transcriptional activity was only observed with higher amounts of Δ40p53 in Saos-2 osteosarcoma cells. Moreover, the level of p21^WAF1^ protein was reduced by expression of Δ40p53 in H1299 cells, whilst it was maintained in Saos-2 cells. Together, these studies indicate that the effects of Δ40p53 on FLp53 activity differ between these cells and that the function of Δ40p53 is highly dependent on its expression relative to FLp53 [20].

#### 4.2.2. Cell-Cycle Regulation

Control over the cell cycle is tightly regulated by FLp53 and Δ40p53. Courtois et al. showed that Δ40p53 is highly expressed at the onset of S phase in serum-starved cells when compared to FLp53 and the isoform becomes predominant up to 30 h after serum stimulation, where its expression was associated with a drop in p21^WAF1^ expression, indicating a transient inactivation of FLp53 functions [15]; and Ray et al. showed that Δ40p53 translation is most active during the G1-S transition, which was accompanied by a transient decrease of FLp53 [59]. These results suggest that Δ40p53 may facilitate G1-S transition by negatively regulating FLp53 expression and activity. A more recent study found that in response to ER stress, FLp53 induces G1 arrest, while Δ40p53 induces G2 arrest by inducing 14-3-3σ [71]. This suggests that, under certain stressors, modulating the ratio of Δ40p53:FLp53 is important for p53-mediated cell-fate outcomes such as the cell-cycle regulation, and that small changes in Δ40p53 expression can have dominant effects on FLp53 activity.

#### 4.2.3. Apoptosis

When Δ40p53 was stably overexpressed the number of dead cells was significantly higher in Δ40p53-lentivirus infected melanoma cells compared with empty vector controls. This was accompanied by a reduction in viability and induction of apoptosis (almost a 5-fold increase), but there were no alterations in cell cycle distribution. These results were only observed in cells which expressed FLp53, suggesting that Δ40p53 cannot induce apoptosis independently of the full-length protein. Activation of p53 by Δ40p53, which is reflected by the phosphorylation of serine 15, was a response to proteotoxic stress caused by the high level of Δ40p53 relative to FLp53. This resulted in increased expression of p53 as well as *PIDD* and suppression of *p21^WAF1^* in melanoma cells [61]. Hence, Δ40p53 expression shifted cell fate away from cell cycle arrest and toward apoptosis [61].

#### 4.2.4. Senescence

It has been demonstrated that Δ40p53 promotes cellular senescence in HCC cells, as shown by an increase in the percentage of senescence-associated β-galactosidase (SA-β-gal)-positive cells and *p21^WAF1^* expression. Moreover, Δ40p53 suppressed clonogenicity and inhibited cell proliferation and survival in the presence or absence of FLp53. Despite an increase in *p21^WAF1^*, there was no increase in *BAX* and *PUMA* expression, caspase-3 and -7 activity, or the percentage of apoptotic cells, suggesting that the suppression of cancer cell growth was not a result of increased apoptosis. However, Δ40p53-induced tumour suppressor activity was attenuated in hot-spot mutant Δ40p53-R175H cells and was reversed following knockdown of Δ40p53. These studies showed that Δ40p53 may have independent functions from FLp53, exerting tumour suppressor activity, rather than simply functioning as an antagonist of FLp53 [59].

### 4.3. Why Is ∆40p53 Functionally Distinct to p53?

A model of one p53 dimer binding to a half-site on the consensus DNA sequence and then further stabilized through binding of a second dimer has been proposed [72], indicating that the transactivation function of p53 is executed through the correct formation of a tetramer. The retention of the oligomerization domain allows Δ40p53 to form homo-oligomers or interact with FLp53 to form hetero-oligomers [14,16,51,73]. Observing the 3D model depicting the interaction of Δ40p53/FLp53 with DNA (Figure 2C), it can be seen that the common loops (TAD II) of these proteins interact with DNA, whereas the loops missing in Δ40p53 (TAD I) are exposed around the periphery, suggesting that a hetero-complex could affect the interactions with p53 cofactors and transcriptional machinery (Figure 2B). Hence, altered or reduced cofactor binding by Δ40p53 may represent one mechanism by which Δ40p53 can alter p53 target gene expression, despite the fact that it contains an intact DNA binding domain and is capable of oligomerisation. 

Due to an absence of the HDM2-binding domain, Δ40p53 may increase the stability of mixed oligomers containing both Δ40p53 and FLp53. Co-transfection of FLp53 with HDM2 in p53-null Saos-2 and H1299 cells, resulted in an almost complete disappearance of FLp53. Interestingly, co-transfection with HDM2 had no effect on Δ40p53 expression levels. Co-transfection of Δ40p53, FLp53 and HDM2 resulted in the maintenance of a detectable level of FLp53 with increased phosphorylation on serine 15 also in the presence of HDM2, which strongly suggests that Δ40p53 is able to protect from HDM2-dependent degradation [20]. A study in Saos-2 cells showed that different gene expression profiles were induced when p53 lacked TAD I, contained non-functional TADs (point mutation of all serine residues) and/or was present in its full-length form. This led to the researchers proposing a model of genes activated by phosphorylation at serine residues in the TAD domain [74], further showing the importance of the TADs in specifying target gene activation by either FLp53 or isoforms that lack the TADs such as Δ40p53.

Δ40p53 is localised predominantly in the cytoplasm, while FLp53 is mostly nuclear. In melanoma cells, Δ40p53 cytosolic localisation was associated with a higher proportion of FLp53 within the cytoplasm, indicating that this isoform likely contributes to p53 mis-localisation within the cytoplasm in melanoma and this is likely to play an important role in defining the function of Δ40p53 [5]. Additionally, p53 mutants show a dominant-negative regulatory effect on FLp53 by converting it into aggregated species, which results in the loss of p53 tumour suppressor roles. In this regard, cytoplasmic Δ40p53 is the major component of p53 amyloid aggregates in endometrial cancer (EC) cells. Transfection of TAD I reduced the aggregation of FLp53, confirming the higher aggregation tendency of Δ40p53, which lacks this domain [68]. Therefore, targeting Δ40p53 specifically may be an effective strategy to prevent formation of p53 amyloid aggregates, thereby reactivating FLp53 function [68].

### 4.4. ∆40p53 in Cancer Prognostication and Clinical Outcomes

Δ40p53 is expressed in normal tissues and cell lines at varying concentrations [14,75]. The expression of Δ40p53 is upregulated in several cancerous tissues and cells including serous ovarian cancer [76], acute lymphoblastic leukaemia (ALL) [77], glioblastoma (GBM) [78], melanoma [5], EC [68], and breast cancer, when compared to normal cells and tissues. Due to its upregulated expression in cancer, it has been suggested that it could potentially be used as a diagnostic or predictive marker [14].

In ovarian cancer, Δ40p53 mRNA expression was analysed in 166 patients subdivided into 91 serous, 30 endometrioid and 45 mucinous ovarian cancer patients by RT-qPCR [76]. This isoform was highly expressed in mucinous specimens when compared to normal ovarian tissue (42 tissues) and its high expression in tumour tissue was positively associated with recurrence-free survival, suggesting that Δ40p53 may indicate a favourable prognosis and could be potentially used as a prognostic marker [76]. However, Δ40p53 expression did not correlate with tumour grade and was not associated with the mutation status of *TP53* [4]. Contrastingly, Bischof et al. [79], using the same method, did not detect Δ40p53 mRNA in uterine serous carcinoma. However, a limited number of patients were included (37 patients) with partial information regarding *TP53* mutation status [79].

Mutations in *TP53* occur in only 3% of primary and 10 to 20% of relapsed B-cell precursor ALL cases. Therefore, it seems that p53 variants could impair the canonical p53 pathway. Recently, Oh et al. [77] investigated p53 isoform expression in marrow from a clinical cohort of 50 patients (40 primary and 10 relapse patients) without mutations in *TP53* by using RT-PCR and western blot. The authors showed that Δ40p53 expression was significantly increased in relapsed patients when compared to control bone marrow cells. Moreover, the authors used a mathematical model to predict the formation of hetero-complexes containing Δ40p53 and FLp53 using their relative expression levels. There was a higher probability of Δ40p53-FLp53 hetero-oligomers in relapse samples than in primary samples, suggesting that Δ40p53 may impair p53 transcriptional activity and that it is associated with the progression of the disease. Even though the small number of healthy controls (*n* = 4) and the heterogeneity of the groups are limitations of this study, their findings present some important insights into the expression of the p53 isoforms in leukaemia [77].

Similar results (to ALL) have been found in breast cancer. Breast cancers exhibit a low frequency of mutations in *TP53* (<25%) and this is dependent on their subtype. *TP53* is mutated in 72% of HER2-enriched and 80% of basal cases, whereas less than 30% of luminal cases harbour *TP53* mutations [80]. It has been evidenced that gene expression signatures in a p53-dysregulated context can more accurately predict chemotherapy responses and outcomes when compared to the mutation status of *TP53* [81,82], indicating that p53 function may be compromised by mechanisms other than DNA mutation. Our research group has evaluated Δ40p53 mRNA expression by RT-qPCR and revealed that this isoform is upregulated in breast cancer and is positively correlated with triple-negative breast cancer—an aggressive breast cancer subtype [69]. We also reported an association between decreased metastasis-free survival and a high Δ40p53:FLp53 ratio [57]. Furthermore, there was a significant negative association of Δ40p53 expression with HER2 positivity. These results support the assumption that p53 isoform levels can be associated with clinicopathological outcomes and that Δ40p53 expression analysis in patients can lead to a different prognostic evaluation [69].

Δ40p53 mRNA expression was also evaluated [83] in renal cell cancer (RCC) of 41 patients with primary RCC and 37 normal adjacent tissues. Its expression was found to be upregulated in RCC patients with *TP53* mutations when compared to FLp53. However, no differences were detected between control and tumour samples, demonstrating that in RCC the *TP53* mutation status could influence Δ40p53 levels independently of the cellular context. Furthermore, no associations were found between high/low Δ40p53 levels and patient survival [83].

Takahashi et al. [78] characterised p53 isoforms in GBM, gliosis, non-tumour brain and neural progenitor cells by using 17 primary human GBM xenografts by SDS-PAGE, immunoblot, mass spectrometry and RT-PCR. Δ40p53 expression was analysed by targeting the intron II sequence of 118 bp unique to Δ40p53 and it was found overexpressed in gliosis and neural progenitor cells when compared to tumours. However, it was not detected in the non-tumour cortex; additionally, Δ40p53 expression was found to decrease with age. These results support the hypothesis that Δ40p53 is associated with regenerative processes and that it may be associated with cancer stem cells. Nevertheless, further analysis of the expression patterns of non-tumour tissues compared with cancer samples could provide new insights as to whether Δ40p53 can be used as a target for controlling cancerous growth [78].

Although mRNA evaluation of p53 isoforms can help to answer quantitative questions, there is a necessity for the development of novel tools to facilitate studies in this area and to predict with accuracy the role that Δ40p53 plays in cancer. The relative mRNA expression evaluation methods require only small amounts of an expressed target, thus making it suitable for studying the low expressed p53 isoforms, but are limited as they only provide a qualitative assessment of their expression levels as either present or absent. Our own studies have measured the relative mRNA expression of the p53 isoforms semi-quantitatively using real-time PCR in breast cancer samples with isoform-specific primers [69]. However, this method is limited by the length of the amplicon. With the exception of one recent study [84] it has not been possible to quantitatively assess FLp53 isoform transcripts, and therefore the contribution of Δ40p53β and Δ40p53γ to cancer phenotypes and correlation with clinical outcomes remains unknown. A major limitation in detecting Δ40p53 at the protein level is the low abundance of the Δ40p53 protein relative to the FLp53 protein and the fact that there are no specific antibodies available for its analysis [20]. Certain isoforms such as p53β and Δ133p53 have commercially available antibodies (TLQ40 and MAP4-9, respectively), making western blotting a reliable method for the detection of β and Δ133 isoforms. However, Δ40p53 can only be detected using a panel of antibodies, where the DO-1 antibody detects epitopes upstream of the ATG40 start site of Δ40p53, therefore detecting FLp53 but not Δ40p53, and 1801 which detects an epitope at 46-55 aa and can detect p53α, Δ40p53, p53β and p53γ. This has made analysis of endogenous Δ40p53 mRNA and protein expression challenging in cell lines and tumour tissues and is likely to be a cause of discrepancies observed between studies. As a result, we still do not have a clear understanding of the expression patterns of Δ40p53 in cancer and what this means clinically. 

Taken together, these studies support both beneficial and detrimental roles of Δ40p53 in cancer in a tissue-specific manner. These outcomes should be interpreted with prudence. Δ40p53 may act in a p53-independent manner [85] or it may also modulate the p53 pathway, where it can substitute FLp53 function, or act as a suppressor or an enhancer of FLp53. Likewise, abnormally expressed Δ40p53 has the ability to switch FLp53 function from growth control to pro-survival and it could be a factor for cancer development and chemotherapy resistance. Additional studies in different tumours with control samples and all isoforms are needed to clarify the complete consequences of Δ40p53 overexpression, its importance in a malfunctioning p53 pathway and the mechanistic implications of its deregulation. Finally, the complex regulation of p53 isoforms must be addressed and understood before they can be used as predictive or prognostic biomarkers.

### 4.5. The Role of ∆40p53 in Aging

Disruption of normal levels of p53 affects its physiological functions, playing a key role in aging and in neurodegenerative diseases [86,87]. Aging is a progressive loss of physiological integrity, resulting in compromised function and augmented susceptibility to death [88]. It is evident that DNA mutation frequencies and tumorigenesis are enhanced throughout aging, which is related to the decline of p53 pathway activity [86]. p53 guarantees the reliability of cell cycle events and responses to stress, thus, mice overexpressing FLp53 are more resistant to cancer but die earlier due to high rates of apoptosis [89,90]. On the other hand, mice expressing low levels of FLp53 develop tumours and consequently die earlier owing to mutations in stem cells and genomic instability [91]. 

Δ40p53 or p44 in mice (owing to its molecular mass of 44-kDa) has been shown to play critical roles in the mouse at all developmental stages. In the developing embryo, p44 is detected well before FLp53 and in embryonic stem cells p44 is required for the maintenance of a highly proliferative pluripotent state [21]. p44 has further been studied due to its potential role in aging and Alzheimer’s disease (AD) [23,92,93]. Maier et al. used transgenic mice overexpressing p44 (p44^+/+^) and observed a progeroid phenotype similar to accelerated aging characterised by memory loss, neurodegeneration, and premature cognitive decline [92]. A high p44:FLp53 ratio led to changes in important ageing factors such as the IGF-1 (insulin-like growth factor 1) signalling pathway [22], as well as developing hypoinsulinemia and glucose intolerance [92,94]. Interestingly, if the mice (p44^+/+^) were crossed with mice lacking endogenous p53 (p53^−/−^), the phenotype was reversed, suggesting that to exert its pro-aging action, p44 requires FLp53, supporting the conclusion that p53 has longevity-assurance activity [95,96,97] and that these proteins act in concert when related to age-associated events in vivo. 

Tau is a microtubule-binding protein that contributes to microtubule stabilization. Analysis of post-mortem human brain specimens and mouse models has shown that aging and memory deficits associated with neurodegenerative diseases are related to the accumulation of hyperphosphorylated tau [98,99,100]. In concordance with the shortened lifespan and increased aging, p44^+/+^ mice also exhibit hyperphosphorylation of tau, synaptic impairment and premature cognitive decline [23]. Pehar et al. also observed that the p44^+/+^ phenotype could be reversed by *Mapt* haploinsufficiency, the gene encoding tau [23], mutations of which have been related to hereditary frontotemporal types of dementia [99]. More recently, the same authors reported that p44 is able to bind to the promoter of numerous tau kinases such as dual-specificity tyrosine-regulated kinase 1a (Dyrk1a), glycogen synthase kinase-3b (GSK3b), and cyclin-dependent kinase 5 (Cdk5) regulatory partners, Cdkp35 and Cdkp39, regulating their transcription and occasioning in amplified mRNA and protein levels [22]. Interestingly, the basal mRNA levels of these kinases were not affected by genetic disruption of *TP53*, suggesting that p44 and p53 are not involved in this regulation. Nevertheless, overexpression of p44 altered the phosphorylation status of tau and the profile of these tau kinases [22]. Additionally, p44 expression was particularly prominent in the nucleus and its levels increased with age in the brain of mice, suggesting that p44:FLp53 ratio imbalance affects tau metabolism and that p44 is activated in an age-dependent manner [22].

Given that aging is the main risk factor for late-onset AD, similar changes, albeit more severe, can be detected in AD patients [88]. A key protein associated with the pathogenesis of AD is the amyloid precursor protein (APP), a type I membrane protein [101,102,103]. AD studies in mouse models show that APP has a causative role in the neuropathology [104,105,106] and *Mapt* downregulation in APP AD models was able to rescue memory loss, implying a promising role of tau linked with APP [107,108]. Therefore, Li et al. investigated the downstream pathway of APP intracellular domain (AICD) and reported that this domain regulates translation of p44 [109]. This study suggests that p44 levels could be a hallmark of age-associated diseases and/or a pharmaceutical target.

It is clear to question whether the role of Δ40p53/p44 in aging and longevity could be also associated with stem cell regulation, given that p53 has been implicated in pluripotent and embryonic stem cell reprogramming [110,111]. p53 functions to maintain the quality and quantity of stem cells by conserving the balance between plasticity and genome stability, which, in the case of impairment, may lead to early aging and the depletion of stem cells. p53 is also able to restrict processes such as dedifferentiation and reprogramming, avoiding the transformation of normal stem cells into cancer stem cells [112]. Similarly, it has been shown in vivo that a high expression of Δ40p53 in early embryos and embryonic stem cells (ESCs) maintains pluripotency and inhibits the more differentiated state. However, in adult somatic cells, high levels of Δ40p53 decrease life span [21,92]. Contrasting results have been found in zebrafish, where its amplified dose impairs overall growth status and leads to developmental malformation of the head, eyes, and somites in zebrafish embryos, further supporting a role for Δ40p53 in differentiation [113].

The role of Δ40p53 in cancer stem cell regulation is poorly understood. Takahashi et al. [78] found a similar pattern to mammalian ESCs, where alteration in Δ40p53 and FLp53 levels switched the proliferative and differentiated state in GBM cells. This finding was associated with the characteristic of GBM being more proliferative when compared to other tissues that express low levels of Δ40p53, corroborating with the theory that this isoform could be considered a neural progenitor marker. Taken together, these studies highlight the emerging role of Δ40p53 in stem cell regulation and aging.

## 5. Can ∆40p53 Be Targeted to Modulate the p53 Pathway?

As indicated throughout this review, ∆40p53 can interact with FLp53 to fine-tune its activity in both a stimulatory and inhibitory fashion. However, this interaction is highly complex and dependent on the ratio between the isoforms, as well as the cellular context [18,20]. With *TP53* being a prototype tumour suppressor gene [114], modulation of its pathway may harbour therapeutic benefits. ∆40p53 may be targeted therapeutically to achieve such modulation, however, this is likely to be complex and case-dependent. Both, strategies to increase and decrease ∆40p53 levels may be desired. While ∆40p53 has been extensively modulated in vitro through transfection, transduction and modulation of G4 formation, only a few agents have been used in vivo and for reasons other than the modulation of ∆40p53. Nonetheless, small molecules may be repurposed to modulate ∆40p53 therapeutically and in vitro approaches may be developed and fine-tuned to suit in vivo needs. Similar to many emerging therapeutics, this approach would be a push towards personalised medicine. The ways by which ∆40p53 may be targeted or has been targeted are summarised in Table 2 and are detailed below.

### 5.1. Modulation of ∆40p53 Using Compounds Already Used Clinically (In Vivo)

Several small molecules that are already used clinically have the potential to alter cellular levels of ∆40p53 and thereby affect p53 pathway activity. While the focus of their current clinical use is not related to ∆40p53, their application in this context is outlined throughout this section.

#### 5.1.1. HDM2 Antagonists

HDM2 antagonists, such as Nutlin-3 [116], may increase steady-state FLp53 expression and alter the ratio between FLp53 and Δ40p53. As mentioned, HDM2 causes degradation of the full-length protein, but not the Δ40p53 isoform, increasing Δ40p53 stability. Additionally, HDM2 may induce p53 translation through an active site that is independent of its ubiquitination site [17]. Thus, inhibition of p53 degradation through HDM2 antagonists can alter the ratio between the two proteins and increase the nuclear levels of both.

This approach may be applicable in instances where ∆40p53:FLp53 is elevated, which is associated with worse prognosis, and *TP53* is not mutated (i.e., FLp53 is not inactivated by another mechanism). An example of this scenario can be found in breast cancer, where *TP53* mutations are present in only a quarter of cases [80], an increased ratio of ∆40p53:FLp53 has been identified in comparison to normal adjacent tissue [69], and this increased ratio was linked to worse disease-free survival [57,69].

With HDM2 antagonists (AMG 232, KRT 232) already being investigated as potential orally administered antineoplastic agents [117], targeting ∆40p53 through this approach may face the least barriers to clinical implementation. Gastrointestinal adverse events may limit the dose of AMG 232 that can be safely administered, but clinical trials have highlighted remissions in a third of patients with AML who retained wt*TP53* (*n* = 13). None of the patients with mutant *TP53* (*n* = 3) responded to AMG 232 [117]. Additional clinical trials are investigating the effectiveness of AMG 232 in metastatic melanoma (NCT02110355), advanced solid tumours or multiple myeloma (NCT01723020 and NCT03031730), brain cancer (NCT03107780), AML (NCT03041688 and NCT04190550), and soft tissue sarcoma (NCT03217266). As data from these trials becomes available, the tolerability and efficacy of AMG 232 in cancer therapy, alone or in combination with existing chemotherapeutic agents and radiation, will hopefully be clarified.

#### 5.1.2. Endoplasmic Reticulum (ER) Stress

∆40p53 expression may be induced through ER stress. Tunicamycin and thapsigargin both cause ER stress through blocking protein folding and ER transit, and non-competitive inhibition of the ER calcium-ATPase respectively. Treatment with these drugs has been found to induce ∆40p53 expression in a cell-specific context [61] as discussed above. With a prodrug of thapsigargin, mipsagargin, having been studied in numerous clinical trials (NCT01777594, NCT02067156, NCT01056029), ER stress induced through these compounds may be exploited to target ∆40p53 therapeutically.

#### 5.1.3. 20S Proteasome Targeting

The 20S proteasome has been found to cleave FLp53, generating ∆40p53 [62]. Small molecules, such as manipulated chlorpromazine, have been found to enhance 20S proteasome activity [118] and may hence increase ∆40p53 levels. As HDM2 antagonists and mipsagargin, chlorpromazine has been and is being investigated in a range of clinical trials (i.e., NCT02943213, NCT04224441) indicating that modulation of ∆40p53 in a clinical setting through this approach may be achievable.

### 5.2. Modulation of ∆40p53 Using In Vitro Strategies

Several research groups have successfully altered ∆40p53 in vitro using an array of transfection and transduction, as well as ionizing radiation. While the practicality of using some of these approaches therapeutically varies due to questions around specificity and safety, others might be developed and fine-tuned into novel treatments modify p53 pathway activity.

#### 5.2.1. Transfection and Transduction for Exogenous Expression

Transfection or transduction of ∆40p53 has been performed in a number of cell lines [5,17,20,21,51,52]. A range of plasmid constructs and viral vectors may be introduced to raise the expression of ∆40p53 or FLp53. Exogenous p53 expression may also be used in the context of *TP53* loss-of-function mutations to provide alternative sources of the protein and its isoforms. 

Site-directed mutagenesis of constructs may replace either the first or the second in-frame start codon with alanine, allowing translation of ∆40p53 or FLp53 respectively [17]. Viral vectors containing the ∆40p53 open-reading frame may also be used to induce expression of the isoform [64,66].

Constructs may also contain hairpin structures near the first start codon, enhancing ∆40p53 expression through the blockage of cap-dependent RNA translation [61]. Masking the first start codon through a hairpin structure was found to accomplish even greater increases in ∆40p53 and constructs containing a stop codon upstream of the second start codon were found to only produce ∆40p53 [61]. Together, these viral constructs indicate that ∆40p53 expression may be increased at desired levels through careful selection of cDNA constructs. Further fine-tuning of exogenous ∆40p53 expression might be achieved by regulating the plasmid concentration used for transfection. Expression of downstream genes in the p53 pathway (*BAX* and *CDKN1A*) indicated dose-dependent effects (both inhibitory and stimulatory). However, this dose-response was not observed when looking at apoptosis, a function thought to be primarily mediated through TAD II [17] and may be difficult to achieve in vivo [17,64,66].

Transduction may also be exploited to eliminate FLp53 expression from one allele. Using adenovirus vectors, Ota et al. deleted exon 2, which harbours the majority of TAD I, and thereby induced ∆40p53 expression. The group was not successful in deleting exon 2 from both alleles [64]. Similar results could be obtained by eliminating the start codon in exon 4 through site-directed mutagenesis of one allele [21].

Increasing ∆40p53 or the ∆40p53:FLp53 ratio may be of therapeutic benefit in cancers where ∆40p53 expression has been associated with improved prognosis [76], or where its increased expression promotes apoptosis [66]. These therapies may even provide an option to treat some patients with mutant *TP53*, as ∆40p53 was found to suppress proliferation in HCC cells regardless of *TP53* mutation status [64]. However, these approaches may carry a risk of permanent adverse effects. Murine studies have indicated reduced lifespan, reduced tissue regeneration, and cognitive decline as possible detriments of ∆40p53 overexpression [23,92]. Thus, transfection/transduction should be limited to neoplastic cells. Targeted transfection may be achieved through intra-tumoural plasmid injection accompanied by focused ultrasound [119]. Similarly, transduction may also be achieved through intra-tumoural vector injection, though in the absence of the site-limiting focused ultrasound more wide-spread adverse effects may be observed. Further in vivo studies are required to determine the suitability of plasmid or vector gene therapy for the modulation of ∆40p53 in cancer. Additionally, future studies may investigate further nucleic acid delivery systems, such as biofunctionalised nanovesicular and polymeric delivery systems [120,121].

#### 5.2.2. Antisense Oligonucleotides

Antisense oligonucleotides are short DNA sequences that can bind to RNA transcripts making them susceptible to degradation by RNase H. Antisense oligonucleotides that target the 5’ terminal region of the p53 transcript were found to limit both FLp53 and ∆40p53 expression. An antisense oligonucleotide (TCCCAGCCCGAACGCAAAG) targeting a sequence upstream of AUG1 at the base of hairpin G56-C169 was found to cause a decrease in both FLp53 and ∆40p53 expression. A different antisense oligonucleotide (AACGTTGTTTTCAGGAAGTAG) targeting a sequence between AUG1 and AUG2 on the atypical loop at the 3’ end of hairpin U180-A218, reduced the expression of FLp53, but raised levels of ∆40p53 in transfected MCF-7 cells. This effect may be the result of mRNA truncation between the two start codons, increasing accessibility to AUG2 [108].

#### 5.2.3. RNA Interference and CRISPR/Cas9

∆40p53 knockdown through RNA interference presents an opportunity to target the p53 isoform produced through alternative splicing of intron 2. Small-interfering RNAs (siRNAs) against intron 2 may partially inhibit ∆40p53, but would not eliminate alternative translation of this isoform through the IRES. However, thus far only FLp53 and isoforms other than ∆40p53 have been targeted by RNA interference [21,64,122] (for a summary of siRNAs used to target p53 isoforms see [123]) and the validity of this approach for ∆40p53 remains to be determined.

Silencing of FLp53 and the ∆40p53 isoform was achieved through CRISPR/Cas9 technology. A single guide RNA (sgRNA) designed to excise a proportion of exon 4 was employed due to a CCT protospacer adjacent motif (PAM) within the exon [64]. Thus far, there is no evidence of CRISPR/Cas9 technology being used to target ∆40p53 alone. While this may be achievable through knockout of intron 2, CRISPR-mediated homology-directed repair of the second start codon in exon 4, or CRISPR interference repressing translation of alternatively spliced p53 mRNA, a lack of PAMs adjacent to these targets or lack of target sequence specificity, may prevent the use of CRISPR technology [124].

#### 5.2.4. Targeting the G-Quadruplex (G4) Structure

Formation of G4 structures within GC-rich regions of intron 3 promotes splicing of intron 2, guiding RNA processing towards fully-spliced p53 transcripts as opposed to ∆40p53 mRNA [52]. This mechanism opens an avenue to target the formation of G4 and thereby gearing the expression of *TP53* towards ∆40p53. Site-directed mutagenesis of the guanines in intron 3 decreased the excision of intron 2 by around 30%, indicating a potential increase in ∆40p53 [52]. Additionally, 320A and PhenDC3—known G4 stabilizers, as well as ionizing radiation, were found to increase expression of fully-spliced p53 mRNA, while ∆40p53 mRNA was decreased [52,53]. As G4 confirmation was most stable in potassium-rich media [52], K^+^ supplementation may also enhance G4 formation and limit retention of intron 2. Sodium ions (Na^+^) may have the opposite effect, being known to destabilise G4 structures [52].

Modulation of G4 formation by the aforementioned mechanisms only had limited effects on ∆40p53 protein levels [52], likely due to increased translation of ∆40p53 through the IRES. Nonetheless, targeting of G4 structures may be used in combination with other mechanisms to target both alternative splicing and translation of p53. Presence of the rs17878362-A2 polymorphism appears to circumvent the necessity of G4 formation for the excision of intron 2 [53], indicating that genotyping may be required prior to applying this modulation strategy.

Additionally, some modulation strategies may increase the sensitivity to existing chemotherapeutic agents and radiation therapy [18]. Beyond their application in cancer, Δ40p53 modulators may be exploited to treat AD and ageing-related disorders, based on the link between p44 and accelerated ageing and neurodegeneration in mice [22,23,92]. However, frequent mutations in *TP53* in cancer, heterogeneity in the response to altered Δ40p53 expression, limited understanding of the physiological function of ∆40p53, and potential side effects may limit the application of these modulators in clinical settings.

## 6. Conclusions

In conclusion, although research on this p53 isoform is continuing, there is still a need to understand the cellular contexts in which ∆40p53 acts in a positive or negative manner. While elevated levels of ∆40p53 have been associated with worse prognosis in some cancers, such as breast cancer [57,69] and ALL [77], ∆40p53 harbours tumour suppressor functions in melanoma [66] and is associated with increased recurrence-free survival in mucinous ovarian cancer [76]. Consequently, molecular profiling of individual cancers, including sequencing of *TP53* to determine mutation status and detection of endogenous levels of both FLp53 and ∆40p53, is necessary prior to altering ∆40p53 levels, if it was to be used as a targeted therapy in cancer or aging related diseases. Such profiling is currently hindered by the lack of commercially available antibodies specific to p53 isoforms, including ∆40p53, and difficulties in distinguishing between isoform and FLp53 transcripts through RNA sequencing approaches. Additionally, p44 has been linked to premature aging and consequently an AD-like phenotype in mice [22,23,92]. While this association remains to be confirmed in humans, it indicates that decreasing ∆40p53 may present a therapeutic option for AD patients. Simultaneously, these findings indicate that premature aging and neurodegeneration may be potential adverse effects of increasing ∆40p53 expression. Hence, modulation of Δ40p53 expression may have beneficial and detrimental effects; and the complex regulation of this p53 isoform must be fully understood before it can be used as a biomarker, or in a therapeutic context.

## Figures and Tables

**Figure 1 cancers-12-01659-f001:**
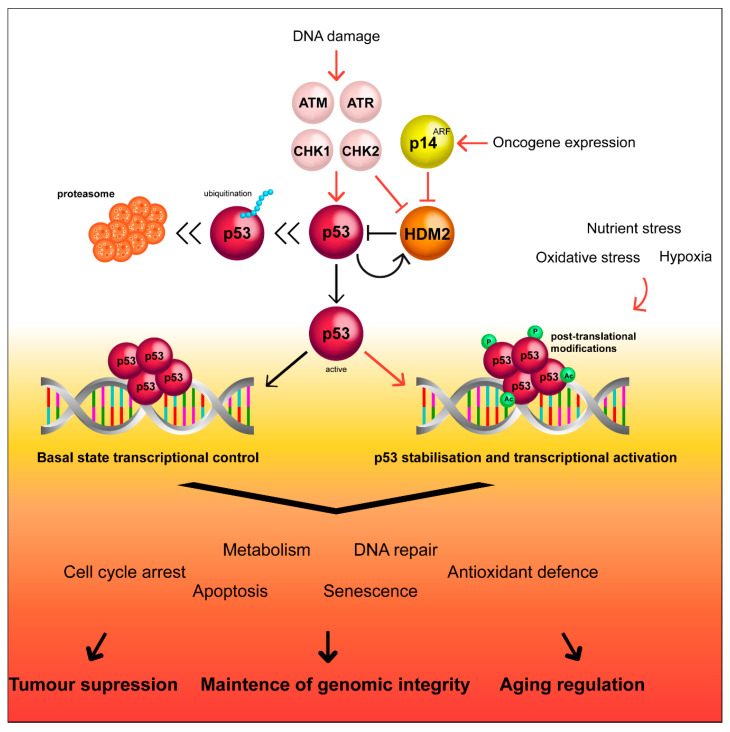
Simplified scheme of the p53 pathway and regulation by HDM2. HDM2 keeps p53 expression levels low, however, upon cellular stress, HDM2 can be regulated by complexing with p14^ARF^, removing HDM2 mediated inhibition of p53. Moreover, kinases can coordinate the p53 response to stressors by inducing post-translational modifications that result in methylation, phosphorylation, acetylation, sumoylation, and ubiquitination of p53 occasioning in stability and transcriptional activation of this protein, maintaining genome integrity.

**Figure 2 cancers-12-01659-f002:**
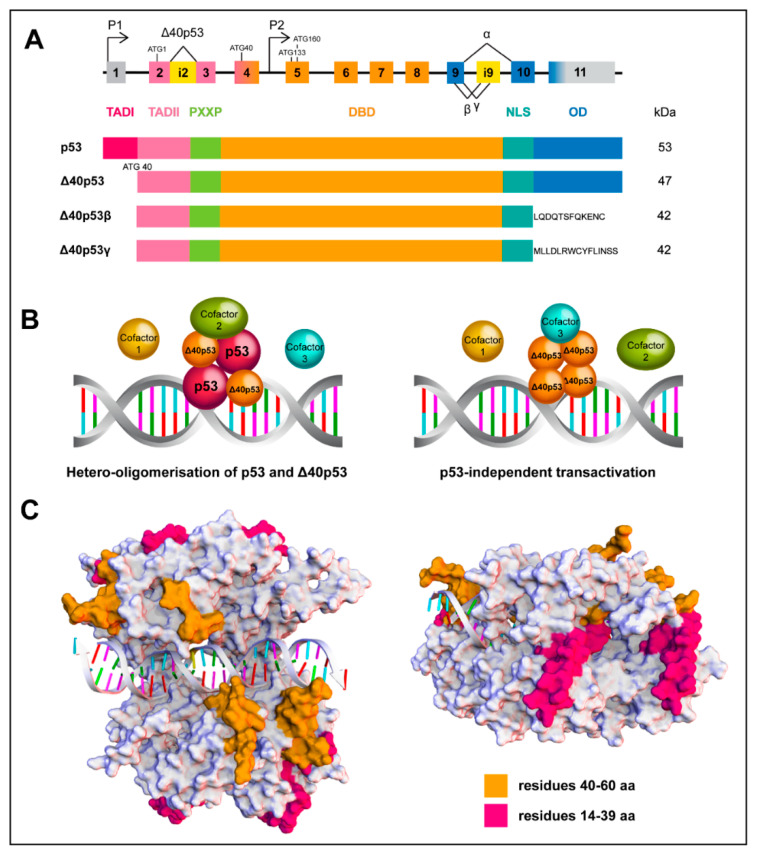
Structure and assembly of wtp53 and the Δ40p53 isoform. (**A**) Scheme of the p53 gene structure; alternative splicing (^) and promoters (P1 and P2) are indicated. Δ40p53 isoform is encoded from P1 due to alternative splicing of intron 2 (i2) or alternative initiation of translation. Partial retention of intron 9 (i9) generates the β and γ isoforms (upper image) Functional domains of FLp53 and Δ40p53 isoforms (lower image). (**B**) Schematic overview of wtp53 and Δ40p53 interactions mediating transactivation. Δ40p53 is capable of forming hetero-oligomers with wtp53 to mediate transactivation (left image) or it can independently mediate transactivation (right image). (**C**) In silico evaluation of differential loops of wtp5*3* and Δ40p53 as a tetramer complex of the DNA binding domain. The common loops (orange, residues 40–60) exhibit close interactions with DNA, whereas the loops missing in Δ40p53 (magenta, residues 14–39) are exposed around the periphery of the tetramer, indicating a possible involvement in protein-protein interactions. The structures of the p53 complexes were generated by homology modelling, target sequence UniProtKB P04637, residues 94–356 and template structures found under the codes: PDB ID 4MZR (Chains A–D, residues 94-358); PDB ID 3EXJ (Chain A, residues 98–291); PDB ID 4IBU (Chains A, B, residues 94–293) and PDB ID 1OLG (Chain A, residues 319–360). Discovery Studio v18.1, Biovia was used, to create the multiple sequence alignments, creation of homology model and loop modelling of residues 14-60 using PDB ID 2K8F (Chain, residues 14–60).

**Figure 3 cancers-12-01659-f003:**
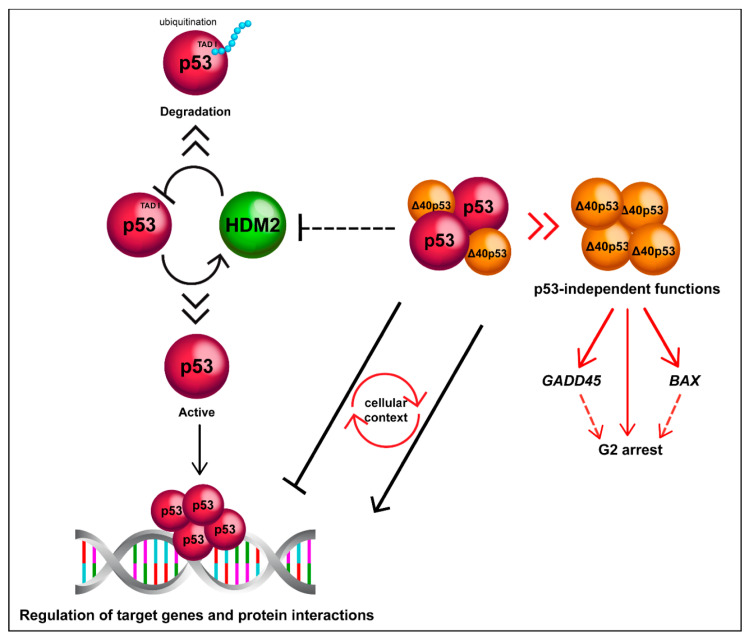
Δ40p53 is functionally distinct to p53. Δ40p53 escapes from degradation mediated by HDM2 and can alter p53 target gene expression in both positive and negative manners depending on the cellular context. Δ40p53 also has p53-independent functions such as transactivation of *BAX* and *GADD45* in p53-null cells and induction of G2 arrest (it is still unclear whether Δ40p53 mediates transactivation as a complex).

**Table 1 cancers-12-01659-t001:** Role of the Δ40p53 isoform in cancer in vitro.

Type of Cancer	Cell Line	Role/Mechanism of Action	Approach Used to Modulate Δ40p53 Expression	Reference
Hepatocellular carcinoma	HuH-1 (*TP53^WT^*), HepG2 (*TP53^WT^*), HEP3B (*TP53^−/−^*, HuH-7 (*TP53^Y220C^*), PLC/PRF/5 (*TP53^R249S^*) human hepatocellular carcinoma cell lines	Tumour suppressor role Promotion of cellular senescence (increase in percentage of SA-β-gal-positive cells) Suppression of clonogenicity Inhibition of cell proliferation Cell cycle arrest at G1 phase Increase in *p21^WAF1^, HDM2, FAS, IL-8* and *GADD45A* expression Increase in protein half-life of *FLp53*	Vector targeted deletion of TP53 exon 2 Retroviral plasmid transduction with constructs containing ORF of Δ40p53 Knockdown of p53 by using retroviral plasmids and CRISPR/Cas9 Inhibition of HDM2 (MG132 inhibitor)	[64]
Melanoma	Mel-FH, Mel-MC, Mel-CV, Mel-RM, MM200, Me1007, Me4405, IgR3, MM186, Mel-KN, Mel-JD, MM283, Mel 4.1, Mel-FC, MM415, MM486, MV3, Sk-Mel-110 and Sk-Mel-28 melanoma cell lines	Expressed at low levels in normal cells but highly expressed in tumour cells Overexpression inhibited transcriptional activation of *PUMA* and *p21^WAF1^* at the basal level and following treatment with DNA damaging agents	Transduction of construct pCIN4.p47 containing the ORF of Δ40p53	[5]
	A375 melanoma cell line	Increase in number of dead cells Reduction of skin cancer cell viability Induction of apoptosis Increase in *p53*, *p21^WAF1^* and *PIDD* gene expression Creation of nuclear tetramers with endogenous, serine 15–phosphorylated *p53*	Transduction with a construct containing the ORF of Δ40p53	[66]
Osteosarcoma	Saos-2 osteosarcoma cell line	Formation of oligomers with p53 Inhibits transcriptional activation of target genes when expressed in excess of FLp53	Transfection using pcDNA3-TAp53 (mut40—ATG40 > TTG) and pcDNA3-Δ40p53 vectors	[20]
Lung cancer	H1299 lung cancer cell line (p53 null)	Formation of oligomers with p53 Suppression of FL-p53 transcriptional activity in an incremental manner Reduction of p21^WAF1^ protein expression when Δ40p53 is overexpressed Enhancement of the pro-apoptotic activity of *p53*	Transfection using pcDNA3-TAp53 (mut40—ATG40 > TTG) and pcDNA3-Δ40p53 vectors	[20]
Colon cancer	HCT116-p53^(–/–)^ (endogenous Δ40p53 expression), HCT116-p53^(+/+)^ (wild-type p53) colon carcinoma cell lines	Enhancement of the pro-apoptotic activity of *p53*	Transfection with adenoviral delivery of a Δ40p53 plasmid	[67]
Endometrial cancer	Ishikawa, RL-95–2, AN3CA and KLE endometrial cancer cell lines	Major component of *p53* amyloid aggregates	None	[68]
Breast cancer	p53 MT cell lines (T47D, MDA-MB-231, HMT-3522-T42) and wtp53 cell lines (ZR-75-1 and MCF7)	Upregulated in tumour cell lines when compared to normal breast epithelial cell lines	None	[69]

*p21^WAF1^*—cyclin-dependent kinase inhibitor 1A; *HDM2*—murine double minute 2; *FAS*—fas cell surface death receptor; *IL-8*—interleukin-8; *GADD45A*—growth arrest and DNA damage inducible alpha, *FL-p53*—full length *p53, PUMA*—p53 upregulated modulator of apoptosis; *PIDD*—p53-induced death domain; *ORF*—open reading frame.

**Table 2 cancers-12-01659-t002:** Mechanisms to modulate Δ40p53 expression.

	Mechanism	Effect on ∆40p53	Effect on FLp53 *	Reference
Targeting alternative splicing (retention of intron 2)	siRNA targeting transcripts that retain intron 2	↓	?	Hypothetical
	Site directed mutagenesis of G in intron 3	↑	↓ (mRNA)	[52]
	stabilisation of G4 through 360A, PhenDC, K^+^	↓	↑ (mRNA)	[52]
	Damage to non-G4 transcripts through ionizing radiation	↓	?	[52]
	Destabilisation of G4 through Na^+^	↑	?	[52]
	CRISPR interference inhibiting translation of p53 transcripts retaining intron 2	↓	?	Hypothetical
	CRISPR/Cas9 knockout of intron 2	↓	?	Hypothetical
Targeting translation via IRES (exon 4)	Transfection with constructs containing a CTG substitution at 2nd ATG	↓	↑ (protein)	[17]
	Transduction with constructs containing ORF of ∆40p53	↑	? [5]; none (protein) in p53-null cells [50]; ↑ (protein) [62,64];	[5,51,64,66]
	Transfection with constructs containing a hairpin structure near/over first ATG	↑	↓ (protein)	[61]
	Transfection with construct containing stop codon upstream of IRES	↑	↓ (protein)	[61]
	Antisense oligonucleotides targeting 5’ end of the p53 transcript	↑/↓	↑/↓ (protein)	[115]
	Tunicamycin or thapsigargin-induced ER stress	↑	↑ (protein)	[61]
	CRISPR/Cas9 homology-directed repair of the 1st or 2nd ATG	↑/↓	?	Hypothetical
	Transfection with constructs containing CTG Substitution at 1st ATG	↑	↓ (protein)	[17]
	Transduction leading to deletion of the second ATG	↑	None (protein)	[21]
	CRISPR/Cas9 excision of exon 4	↓	↓ (protein)	[64]
Other	Nutlin-3 and MG132 inhibition of HDM2 ubiquitination of FLp53	↑;↑FLp53:∆40p53	↑ (protein)	[17,64,116]
	Transduction leading to deletion of exon 2	↑	?	[64]
	20S proteasome enhancement through manipulated chlorpromazine	↑	?	Hypothetical

* Effects on FLp53 are only included if measured by the cited study. Effects on FLp53 are likely to be complex and difficult to predict as both the ∆40p53 modulation strategy and the change in ∆40p53 may affect FLp53 levels; and effects on FLp53 may be cell- and context-dependent. ↓—decreased expression; ↑—increased expression; CRISPR—clustered regularly interspaced short palindromic repeats; ER—endoplasmic reticulum; G4—guanidine quadruplex; IRES—internal ribosome entry site; K^+^—potassium ion; HDM2—human double minute 2 homolog Na^+^—sodium ion; ORF—open reading frame; siRNA—short interfering RNA.

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
