# Peer review of "Good Cop, Bad Cop: Defining the Roles of Δ40p53 in Cancer and Aging"

_cancers, 2020, doi:10.3390/cancers12061659_

Round 1
Reviewer 1 Report
This manuscript focuses its attention on one of the isoforms that are produced by the alternative splicing of TP53 gene. In the recent years, the importance of these isoforms in the control of cellular homeostasis has emerged as each isoform seems to modulate different pathways assigned to the response of cells to different stimuli.
In the scientific literature, a review describing and summarizing in detail the functions of the Δ40p53 isoform is missing. In several publications Δ40p53 isoform has been seen to be involved in aging and cancer, but it could have other debated functions. However, its role appears controversial. So the choice of the authors to stress this isoform confers novelty on the manuscript.
This review is very comprehensive, it really addressed the most recent works in literature and the functions of Δ40p53 known to date.
The figures and tables are also appropriate.
I really have no criticism to make. I recommend the acceptance to Cancers.
Author Response
We thank the reviewer for their comments and support.

Reviewer 2 Report
The review entitled “Good cop, bad cop: defining the roles of ∆40p53 in cancer and aging” by Reinhardt et al, clearly summarizes what is known about the ∆40p53 isoform, from production of the protein to expression in cancer, including potential therapeutic opportunity. This review highlights the conflicting function of ∆40p53, either an activator or inhibitor of FLp53 as well as an independent protein.
Although this review is complete, some aspects are lacking for the readers to have an entire view of the complexity of data interpretation when speaking of these particular isoforms whose expression are intimately related due to TP53 gene organisation.
Major comments
- line 60: the indicated percentage of TP53 mutation is extremely high (i.e. 80%). Most of the papers including the IARC TP53 database usually speak about 50%, the second half being mainly inactivation by viral and cellular protein interaction. Does it encompass both somatic and germinal mutations? This sentence should be carefully verified and written.
- lines 69-85: a sentence should be included to emphasize that the molecular mechanism(s) by which p53 “chooses” its target genes depending on the stress intensity has been extensively studied but still remains to be clarified. Few examples should help in depicting the actual vision of such regulation and illustrate the importance of p53 isoforms in understanding this key point when speaking about p53 pathway.
- line 137-139: since this mode of production of ∆40p53 is relatively new, it should be explain in more details. Does it correspond to the hypothesis raised by Courtois et al, Oncogene 2002 about production of ∆40p53 from FLp53 through calpain cleavage? In addition, to facilitate the reading of ∆40p53 production, one suggestion might be to start from splicing, translation then post-translational modification.
- Table 1: It would be useful to indicate the approach used to modulation ∆40p53 expression for each studies. In addition, general sentence should be avoid (e.g. “important role in…”) and specific sentence such as “promotion of…, inhibition of…” should be preferred.
- Related to the previous point, line 396: Paragraph on “∆40p53 targeting to modulate p53 pathway”. Are these example given for in vitro/in vivo studies or therapeutic treatments? It is not clear and there is a mix of both approaches that should be separated to (1) better appreciate the limit of the conclusion from biological studies and (2) focus only on global approach that can be used in humass. One suggestion: a section about FLp53/∆40p53 modulation in vitro before molecular/biological function (that will help in introduce limit in these studies) and a last section about ∆40p53:FLp53 targeting in cancers.
- Table 2: add potential effect of the different ∆40p53 targeting on FLp53 since modulating ∆40p53 will intimately affect FLp53 expression. This point is key to show the limit and difficulty in studying FLp53/∆40p53 biological function.
- Paragraph 4.4: It is essential to indicate the tools used to analyse ∆40p53 expression (example: RT-qPCR targeting intron 2-containing FLp53) and indicate the limit of detection in human samples. This point should be taken into consideration when coming to conclusions. It may explain contradictory data because, to date, there is no a clear vision of ∆40p53 expression in human cancers. This point should be discussed.
Minor comments
- line 51-53: references are lacking
- Figure 2: (A) TP53 gene organization is lacking to fully understand the generation of ∆40p53 isoform. (B) What about competition of FLp53 and ∆40p53 tetramers to bind p53-RE? (C) Legend should be directly added on the figure to directly understand it (in particular magenta colour)
- lines 111-113: What is the utility of this sentence regarding the following one? Is it to highlight that in response to stress, while an overall decrease in protein synthesis occurs, the presence of IRES in TP53 mRNA favours the translation of FLp53 and ∆40p53 thus allow to establish a cellular response?
- lines 125-128: reference is lacking
- A sentence should be added in the introduction part to indicate that most of the studied present here focus on ∆40p53a and that few (if nothing) is known about ∆40p53b and ∆40p53g.
- 6.line 228: “distinct”
- line 369: remembering what AD means could help the reader
Reviewer 3 Report
In my opinion, this is a very well written and almost complete manuscript. The figures are very well prepared, and the authors are proficient in preparing scientific works. I feel that the introduction should include more information regarding the role of p53 in inflammation, a hallmark of cancer (e.g PMID: 30344124 ). Furthermore, the authors may wish to reduce the number of the outdated references, unless they believe are necessary.
Reviewer 4 Report
In this review by Luiza Steffens Reinhardt et al, currently understood and studied roles of
Δ40p53 in cancer and aging was thoroughly discussed. Despite the ambivalence of Δ40p53 in its functions, the paper is straightforward, and well-organized with tables and illustrations to guide the readers through. The paper deserves to be published and is a valuable contribution to the “cancers” journal. Some minor flaws need to be addressed before publication.
Major revision:
- Why the authors choose Δ40p53 isoform?
“The remainder of this review will focus on Δ40p53.” page 4/25, line 106-107:
After this last sentence, it would be better clarified if the author added in using few sentences to explain why Δ40p53 is reviewed amongst the other protein isoforms of p53. It would enhance the flow of the paper.
- Please explain if Δ40p53 is a natural isoform.
- According to your explanation, Δ40p53 serves both positive and negative functions. Then, how is the expression of Δ40p53 correlated to tumorigenesis in several cancers?
Minor Comment:
- “4.2.1. Modulation of transcription of p53-target genes”: page 9/25, line 197-199:
For the sentence "Moreover, the level of p21WAF1 protein...”, as the last sentence of the paragraph, it would be better to add a phrase that emphasizes about ratio-dependent function of Δ40p53 on FLp53, in addition to cell to cell differences.
- “4.2.2. Cell cycle regulation”: page 9/25, line 201-203:
Although the first sentence states “Control of the cell cycle is tightly regulated by FLp53 and Δ40p53”, there is a lack of specific mechanism of cell cycle control within the paragraph itself. For example, high expression of Δ40p53 onset of S phase (line 202) and its peak activity during G1-S transition (line 203) does not further explain about what the results of increased Δ40p53 mean. It would be better to mention briefly about the results from the references [14] and [48].
- “4.2.2. Cell cycle regulation”: page 9/25, line 204-206:
The word ‘outcomes’ (line 206) in the sentence “This suggests that, under certain stressors…” remains vague. It would clarify the meaning of the paragraph if outcomes were elaborated.
- Typo-error: page 10/25, line 228:
In the subheading, “Distict” should be corrected as “distinct”.
- “4.3 Why is Δ40p53 functionally distinct to p53?”: page 10/25, line 228:
The final sentence “Thus explaining how Δ40p53 can alter… in a positive and negative manner” fails to well-establish the connection between the overall flow of the paragraph in lines 229-238. Although there is an interaction between Δ40p53/FLp53, such description is not enough to correlate with control of p53 by Δ40p53 in a positive and negative manner. More details are required or the final sentence should be improved to enhance the link between Δ40p53/FLp53 and 40p53’s control of p53.
- Is Δ40p53 isoform not related to DNA repair?
Reviewer 5 Report
The aim of the present review by Reinhardt et al. was to summarize the current knowledge on the role of N-terminal truncated p53 isoform delta40p53 in cancer and aging.
In my opinion as a peer reviewer, this is an intelligently written, intriguing review following a clear logical flow. The authors may consider adding a chapter where to discuss the connection between aging and cancer and to highlight the role of p53.
Reviewer 6 Report
This review focused on the role of delta40p53 in cancer and aging. The authors have cited appropriate papers and I think they fully explain the function of delta40p53. Therefore, I think that this review is suitable for publication in Cancers.
My specific comment; The title "Good cop, bad cop" is thought to be inspired by DePinho's review article "p53: good cop/bad cop" (Cell, 2002), but the authors have not cited this review article. I think that the authors should cite this review to explain the significance of this title.
Round 2
Reviewer 2 Report
The authors answered all the comments that have been raised.